# Parent-Child Relationship in the Civil Code of China

**Wenting You**

School of Law, Shanxi University, Taiyuan 030006, China; youwenting@163.com

**Abstract:** The purpose of this article is to familiarize readers with the Chinese Civil Code, which entered into force in early 2021, and to draw their attention to the changes brought about by the Marriage and Family Book, which is now included in Volume V of the new code. The paternity system best reflects the changes in the Chinese Marriage and Family Book, especially Article 1073. A complete paternity system includes presumption, claim, and denial of the parent-child relationship. However, Article 1073 of the Civil Code, which regulates the parent-child relationship, is a guiding provision with a lack of operational rules. It is necessary to make general rules for operation and enforcement by adding supporting rules, including the presumption of legitimate children, the claim of children born out of wedlock, the denial of legitimate children, and other operational rules, to resolve paternity disputes. The Civil Code also makes changes to the adoption system in the Marriage and Family Book, mainly by further restricting the conditions for adopters with the aim of protecting the interests of the adoptee children. Although the Chinese Civil Code retains the concepts of legitimate and illegitimate children, in essence, there is no difference in their rights and legal status, including the right to inheritance. In conclusion, the legislative norms of paternity determination improve the Chinese paternity system, but lack operability, and it is important to accumulate experience through practice and draw on custom and jurisprudence to develop specific operational rules that complement the legislative provisions. This is exactly what this paper will address and the knowledge gap it will fill.

**Keywords:** marriage and family law; private law; parent-child relationship; presumption; adoption; denial of legitimate children

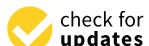



## 1. Introduction

The Chinese Civil Code strives to include all current legal rules in the realm of civil affairs into its 1260 items in a methodical manner. It is a codification and revision of Chinese civil legal norms, in the same structure as the Swiss, German, and Japanese civil codes. It is split into seven volumes: General Provisions, Property, Contracts, Personality Rights, Marriage and Family, Succession, and Torts. The 2020 codification is a recombination of existing minor acts, a compilation that does not change the fundamentals of present regulations. It is neither a reinterpretation of existing texts nor the establishment of new private law, but rather the integration of pre-existing laws.

Paternity is a newly created institution that represents a significant development in Chinese family law. It is included in Article 1073 of the Civil Code, which represents the paternity legal system in Chinese civil law. The recognition or denial of paternity by parents and the recognition of paternity by adult children are consolidated in this article of the Chinese Civil Code. Even though there is only one article, it establishes the legal framework for parentage recognition in China and represents a significant step forward in the development of Chinese kinship law.

Paternity confirmation and denial are both included in the text. The first is paternity confirmation, often known as the claim of children born out of wedlock, in which the biological father acknowledges and treats the child born out of marriage as his child. On the other hand, there is paternity denial. It is the disidentification of a child born in

marriage by their father or mother, who provides negative proof, which either overturns the evidence-based system of presumed wedlock or denies that the child is born in wedlock. However, it remains in theory and must be reinforced with practical rules to move implementation forward.

## 2. Parent-Child Relationship: An Overview

This section presents a brief review of critical methods for understanding the parent-child relationship in the literature, including law, history, economics, politics, and traditions.

Parent-child relationship legislation has undergone a refinement process. The term "parenthood" is derived from the Old Roman phrase "patria potestas" (N. Zhou 2001), which refers to the parent-child connection being totally subordinate to the family system, which is administered by family law and marked by patrilineality, patriarchy, and paternal authority. Later, on the principle of equality between men and women, the parentage emphasized both parents' parental power over their children. In modern times, the parent-child connection is concentrated on the protection and raising of children, and parental power is only recognized for the performance of the commitment to protecting and raising children. There is a trend toward "Best for children" in the parent-child legal relationship, and the power of parents has been weakened.

States have gradually shifted the focus of parenthood laws to children's rights, emphasizing the duties and responsibilities of parents towards their children. The genuine "child-oriented" parenting law promotes children's interests, the protection of children's rights, children's education, and parents' duties and obligations. The basic concept, purpose, and task of the parentage law in the future civil code should focus on the children's best interests, pay attention to the protection and education of minor children, and emphasize the duties and responsibilities of parents. In the Civil Code, the primary idea, goal, and mission of parentage law should focus on the best interests of children, the protection and education of minor children, and the obligations and responsibilities of parents, i.e., "child-based" legislation.

The study of parentage law in China has gone through three stages: First, the 1980s were the starting point, and the primary foundation was the seven clauses of the 1980 Marriage Law concerning the content of parenthood. Second, the 1990s were a period of progress when a study on parentage legislation gradually began until the original Marriage Law of China was amended in 2001. The study includes not only the determination of paternity but also a discussion of its philosophy. Third, the twenty-first century, the stage of expansion and depth. The legal system for determining paternity was adopted into the Civil Code and quickly gained recognition.

The issue of parent-child relationships has been widely discussed. Scholars' primary points of view on the paternity system revolve around the presumption (Zhao 2016) and denial systems of legitimate offspring (Wang and Meng 2020), illegitimate children (Chen 2020b), claims (You 2021), and legitimation of children born out of wedlock (X. Zhang 2018), mainly concerning the systems of presumption and denial of legitimate children and the systems of claim and legitimation of illegitimate children (Long and Feng 2022). There are some arguments in favor of such separate provisions, and at the same time, some argue that there is no claim system in Chinese current law and that the establishment of a claim system should take due consideration of the blood relationship, the claimant, the legal period, and the conditions to be met for a voluntary and compulsory claim (Luo 2018). Some think the system should be established with due consideration for the blood relationship, the claimant, the legal period, and the conditions to be met for voluntary and compulsory claims (H. Zhang 2015). It is also believed that the current social development no longer distinguishes between legitimate and illegitimate births. Therefore, there is no room for the application of illegitimate children recognition, and it should be unified (Xue 2014). Some scholars believe that because foreign laws have abolished the title of illegitimate children, it is possible to learn from them and no longer divide the identification of children, but to

legislate from the perspective of parents and establish a system of paternity recognition and paternity confirmation (Huang 2006).

The scholars have also discussed the definition and nature of parental authority (Feng 2014), the difference between parental authority and guardianship, the exercise and protection of parental authority (Xia 2016), and other issues. Regarding the parental authority (parental care) system, there is a dispute about whether to adopt the notion of parental authority or take it from foreign laws such as German law (Chen and Li 2017). The meaning of parental authority is indeed controversial, as is whether the concept of parental authority or the concept of parental care in foreign law, such as German law, may better reflect parents' obligations to their children. There was even a debate about the nature and definition of parental authority, and the relationship between parental authority and custody (Xiao 2017).

Although the new Chinese Civil Code incorporates parentage norms, it still maintains the "grand guardianship" legislative model of incorporating parental authority into the guardianship system from a stylistic point of view (Jiang and Han 2001). Chinese kinship law does not introduce the concept of parental authority, which has an entirely different social function from guardianship. Parental authority has the role of protecting the interests of minors and preserving the typical order of family life. At the same time, guardianship is designed to compensate for the lack of capacity of juveniles and adults and to protect their interests. It is clear that, in terms of legislation, parental authority and guardianship should be legislated separately. The replacement of the parental authority system by the guardianship system in Chinese current parentage law is a system of purely social obligations based on social relations based on blood and kinship, which equates the love between parents and children. This is natural to human beings and deepened by Chinese traditional culture, with general social obligations, and is essentially a deficiency in the legal system (Ma 2008).

It is also ineffective in regulating parent-child relationships in China today. So, the innovation should be adopted in the new civil code to keep up with the times. As such, a systematic review of the literature has been conducted. The establishment of the paternity system in the Civil Code through Article 1073 is an important manifestation of the improvement of Chinese kinship law, although it still has some shortcomings. However, this is exactly what we will explore in this article.

## 3. The Chinese Parent-Child Relationship Laws

Unlike civil tradition, the Chinese Civil Code divides family law into two separate directions, one on marriage and the other on inheritance. The Marriage and Family book combines the general theory of the parent-child relationship with the various judicial interpretations formulated by the People's Supreme Court. Several Chinese legal academics of marriage adopted the legislation suggestions from the society and urged for the establishment of complete legal institutions to affirm the parent-child connection, which contains three components: the presumption of parentage, the claim, and the denial of the parent-child relationship (Dan 2019).

Let us look into Article 1073 on the parent-child relationship stipulated in Volume V. Volume V is mainly based on the 1980 Marriage Law (with amendments in 2001) and the judicial interpretation of the Supreme People's Court of China. Before the NPC Deputies' Congress voted on the Marriage Law, there was no regulation about the presumption of a parent-child relationship. Article 1073 of the new Civil Code incorporates this judicial interpretation. This is undoubtedly an essential improvement over family law, although it has been adopted in case law. The original Marriage Law did not deal with the paternity of confirming and denying, leading to a loophole in the legislation.

Article 1073, with two items, stipulates that:

Where an objection to maternity or paternity is justifiably raised, the father or mother may institute an action in the people's court for affirmation or denial of the maternity or paternity.

>   Where an objection to maternity or paternity is justifiably raised, a child of full age may institute an action in the people's court for a determination of the maternity or paternity.

Article 1073 clarifies the confirmation or denial of a parent-child relationship. One of the fundamental institutional aspects of the civil code is the parent-child connection, which is critical to the family. Legally, the parent-child connection refers to the relationship between parents and children, including their rights and duties. As Frederick Engels observed in his book Origins of the Family, Private Property, and stated that (Engels 2010):

>   The names of father, child, brother, sister are no mere complimentary forms of address; they involve quite definite and very serious mutual obligations which together make up an essential part of the social constitution of the peoples in question.

The parent-child connection must be legally recognized before their rights and duties may be created.

Chinese law is in line with other national legislation in terms of the identification of motherhood. The legislation of most civil law countries follows the constant principle of Roman law that "the one who gives birth is the mother", and that the fact of birth automatically gives rise to motherhood. As a rule, the legal relationship between the child and the biological mother is not easily disputed, and motherhood is automatically acquired on the basis of the fact of the child's birth or the registration of the mother's name on the birth certificate. The acquisition of paternity or maternity is linked to the existence of a marital relationship and the fact of the birth of the child. The presumption of motherhood is generally based on the fact of birth, registration, or possession of identity. In addition, motherhood can also be established by possession of civil status. Take French law for example, Article 311-1 of the French Civil Code provides that, in addition to the fact of birth, motherhood may also be established by possession.

The mother-child relationship can be acknowledged on the basis of birth rather than through legal channels; however, the father-child relationship is more complicated, owing to the fact that biological fathers are not always present. The legal presumption of paternity is used in most countries. A child conceived or born during a wife's marriage is presumed to be her husband's child, and the father-child relationship is legally recognized. For example, Article 255 of the Swiss Civil Code:

>   1. Child born during the marriage is presumed to be the husband and wife's child;
>   2. If a baby is delivered within three hundred days after the husband's death, or three hundred days later but with proof that the baby was conceived before the husband's death, the husband is legally believed to be the father in both situations; 3. If the husband is declared missing by laws, within three hundred days from the date of the danger happening or the last hearing, the husband is legally presumed to be their father.

In China, there was no law requiring parental consent until the 2020 code.

They are based on the legal presumption of parentage instead of the provision. Article 1073 of the Chinese Civil Code covers both the acknowledgment and the denial of paternity. One is paternity confirmation, also known as the adoption of children born out of wedlock, which refers to the act of the biological father recognizing the child born out of wedlock as the father and treating it as his child. The other one is paternity denial. It is the disidentification of a child born in wedlock by their father or mother providing negative evidence, which overturns the system of presumed wedlock proved by the evidence or denies that it is a child born in wedlock.

The Chinese Civil Code retains the title of children born out of wedlock; however, these children are treated no differently from legitimate children in the Civil Code, and they have exactly the same rights and obligations. In other words, children born out of wedlock enjoy the same rights and obligations as children born in wedlock, such as the right to be supported and the right to inherit from their parents' estates, without any difference.

This explains why the Chinese Civil Code does not provide for a system of claim and legitimation, since children born out of wedlock have the same legal status as those born in wedlock, even if they do not become legitimate children through adoption and legitimation.

The legal identity of parents is the presumption of a parent-child relationship. A baby is a child of both couples based on the principle of equality between husband and wife and the reality that pregnancy and childbearing are borne unilaterally by the mother. This paternity presumption verifies not just the father but, more significantly, the mother (Xue 2018). As a result, the provisions on objection to parent-child relations explicitly state that both parents have the right to request confirmation or denial of the parent-child relationship and do not consider the traditional notions of denial of children born in wedlock or adoption of children born out of wedlock.

The code defines denial as the rejection of a child conceived or born in a legally recognized parent-child relationship. A parent may file a denial lawsuit if they can demonstrate that the father or mother has no biological relationship with the child. The right to paternity confirmation is the legal recognition that an individual is the biological father or mother of a child. A paternity confirmation action may be filed in the Primary People's Court by any father, mother, or adult child who can demonstrate the blood relationship between the children and their parents. In addition, the code only permits adult children to sue for paternity confirmation and not denial. This is done to prevent adult children from evading their responsibility to care for their aging parents. While pursuing the authenticity of blood relations, objections to parent-child relationships must also consider the stability of parent-child relationships and safeguard the legitimate rights and interests of elderly individuals.

The establishment of the paternity system in the Civil Code through Article 1073 is an important manifestation of the improvement of Chinese kinship law, although it still has some shortcomings. This deficiency can be remedied through theoretical debate and judicial interpretation, which do not affect the progressive nature of the Chinese Civil Code.

## 4. Presumption of the Parent-Child Relationship

Although Volume V does not use presumption to confirm or deny parent-child relationships, it has established the essential criteria for raising objections to parent-child relationships based on judicial practice experience, which provides guidelines for dealing with problems of parent-child relationship confirmation and denial in practice.

### 4.1. Legal Nature

4.1.1. Civil Law Presumption Norms

In civil law, what is a presumption? Presumptions, including the underlying facts and presumed facts, are the parties' responsibility to prove the underlying facts in order to obtain legal confirmation of the presumed facts and produce particular legal effects. Adjudication is defined by the adjudicator's process of making assumptions. The parties present evidence of the underlying facts in order for the arbitrator to make factual determinations and then legal presumptions. The application of legal standards and the allocation of parties' interests may vary or be incorrect if the identification of facts deviates. Consequently, when the solid state is unknown, the referee must rely on the burden of proof to evaluate reality in order to apply the law, and the referee is responsible for this process of presumption. The presumption principle connects fact to law.

Presumed norms are credible, mandatory, and refutable. Credibility, the result of scientific presumption norms, means that after the party asserting the presumption has presented evidence to prove the underlying facts and the other party has not presented rebuttal evidence, the adjudicator may simply find the presumed facts. Mandatory, after the facts are clear and proven, relying on legal norms directly presumed and found to be accurate, the judge does not make presumptions based on his judgment determinations he does not need, and cannot free his mind from evidence. After the party asserting the presumption shows the underlying facts, the opposing party may rebut the presumption, either by

testifying against the underlying facts supporting the assumption or by demonstrating the supposed facts themselves to be false, to invalidate the presumed facts.

### 4.1.2. Presumption of Parent-Child Relationship Norms

Based on their nature, presumptions are divided into legal and factual presumptions. The presumption of paternity is substantial because it is based on the principles of life experience, but it is also a legal presumption as established in the Marriage and Family Book and its interpretations. Whether it is a presumption of fact or a presumption of law, presumptions are essentially a way for adjudicators to assign points when the truth or falsity of the facts is unknown. In the absence of evidence to prove the existence or nonexistence of paternity, the adjudicator determines paternity based on a combination of factual and legal presumptions, with the intention of protecting the interests of the parties and society based on paternity.

The presumption is based on the concept of refutability, except that it cannot be refuted. A presumption is a probability-based logical rule, whereas an irrefutable premise exists only in exceptional circumstances. In the United States, for example, the California Evidence Code of 1965, Section 621, defines legitimate children, and a child born to a wife living with her husband is a legitimate child so long as the husband is able to accept children. This presumption of paternity is irrefutable primarily because the regulations on paternity in various countries and regions, such as the United States, will reflect their social policies and inclination to want to protect their interests, and because the presumption of paternity is social, it is crucial that it be stable and irrefutable.

Article 39 of the Interpretation (I) of the Supreme People's Court on the Application of the "Marriage and Family" Book of the Civil Code of the People's Republic of China (Interpretation I of Marriage and Family Book) stipulates that the legal presumption of paternity is rebuttable. The adverse presumption resulting from the rejection of a paternity test is theoretically high, but the primary purpose is to protect the interested party's right to know the existence of fatherhood and to guarantee the validity of the lineage, in addition to the stability of paternity. According to this rule, paternity is presumed to exist when one party claims to confirm fatherhood, the other party declines a paternity test, and there is no contrary evidence. If additional evidence indicates that the child's biological father is someone else, the current paternity may be reversed in order to validate the lineage.

### 4.2. Legal Logic

Legal logic is the logic reasoning that leads to a specific legal conclusion. There are three possible logical relationships between the underlying facts and the existence of paternity: causation, incompatibility, and subordination. Through these three relationships, the adjudicator can make an accurate presumption that the parties' conduct is in accordance with the law. The general components of legal logic are the main premise, the minor premise, and the conclusion. Adjudication is the process of reasoning and concluding using legal logic, where the central premise is the legal provisions and daily rules of thumb, the minor premise is the determination of the facts, and the conclusion is the outcome of adjudication (Li 2017).

According to Article 1070 of the Civil Code, "parents and children have the right to inherit from each other." According to the provision, Tom has the right to inherit Harry's estate if he is Harry's son. In this example, the provisions of the Civil Code are the major premise, while the underlying fact that Tom and Harry are father and son is the minor premise. Harry can inherit from his father Tom because the minor premise is consistent with the major premise, i.e., the underlying facts are legal. Evidently, the assumption is the means by which the underlying facts are deduced using legal logic.

The new civil code's presumption of paternity is a legal presumption that must adhere to the rules of logical reasoning. The legal logic of the presumption of paternity is the process by which the adjudicator reasons from the facts underlying the parent-child relationship to the presumed reality of paternity. Specifically, the central premise



is the provisions of Marriage and Family Book Interpretation I, the minor premise is the underlying facts, and the conclusion is that paternity exists or does not exist.

In paternity presumption, the presumed fact is whether paternity exists, and the presumption process is straightforward, whereas determining the underlying facts is more difficult. According to Interpretation I of the Marriage and Family Book, there are three conditions for the presumption of paternity: first, one party must have evidence proving the existence or nonexistence of paternity; second, the other party must have no evidence to the contrary; and third, the other party must refuse to take the paternity test. This legal provision closely resembles the underlying circumstances in form. Nonetheless, relying solely on these three conditions as the central premise of the logical assumption may result in an incorrect conclusion. Even if one party has evidence that paternity does not exist, the other party does not refute it, and there is no paternity test, it is not necessarily the case that paternity does not exist. This legal presumption is merely a legal framework for allocating the burden of proof in the paternity assumption; it does not pertain to the underlying facts. The factual basis for the presumption of paternity should be whether both parents have created a parent-child relationship with the child.

The facts underlying the presumption of paternity must be determined precisely. Paternity is presumed to exist if the woman engaged in sexual activity with the alleged father during gestation. Paternity may be presumed absent if the underlying facts are that the man did not have sexual relations with the woman despite the fact that they lived together during her conception (for example, because he was away on business), that the man was infertile, that the woman had sexual relations with other men during her conception, etc. If paternity does not exist because the woman has not had sex with the man, the man is infertile, or the person who has had sex with the woman is not a party, and there is no evidence to the contrary and no verification of paternity test results.

*4.3. Legal Effects*

The legal impact of the assumption is a change in the parties' rights and responsibilities following the presumption. It manifests itself in both procedural and consequential impacts in the parent-child interaction.

4.3.1. Procedural Impacts

The distribution of the burden of proof influences the establishment of the facts. Because this rule of assigning the burden of proof objectively reduces the difficulty of proof for the party with the burden of proof and alters the distribution of procedural rights and obligations in litigation, which in turn alters the configuration of substantive rights and obligations, presumption can determine the facts when the truth of the facts to be proved is unknown Ibid. (Xue 2014). The procedural impact of the presumption of paternity is to transform what would otherwise be a problematic abstract proof into a less difficult and more concrete one (Zhu et al. 2019). The presumption of paternity changed the plaintiff's object of proof, shifting the plaintiff's proof from the existence or non-existence of paternity to being able to infer the proof of the underlying facts of paternity's existence, both in theory and in practice, the proof of the underlying facts being much less difficult than the proof of the presumed outcome. The presence or non-existence of paternity is abstract. However, the fundamental reality of whether a woman has sexual relations with her father during conception is relatively difficult to prove.

The presumption of paternity reduces the burden of proof for the plaintiff. The burden of proof is essentially the risk of prejudice to the claiming party when the truth of the facts is unknown, and this modification unquestionably reduces the plaintiff's burden of proof. However, this burden is not eliminated, only reduced, and the plaintiff still risks losing the case if they are unable to prove the underlying facts.

4.3.2. Consequential Impacts

Article 39 of Interpretation (I) of the Supreme People's Court on the Application of the "Marriage and Family" Book of the Civil Code stipulates that:

Where the father or mother institutes an action in the people's court for denial of the maternity or paternity and has provided necessary evidence, if the other party has no contrary evidence and refuses a parentage test, the people's court may determine that the claim of the party denying the maternity or paternity is tenable.

Where the father or mother or a child of full age institutes an action for confirmation of the maternity or paternity and has provided necessary evidence, if the other party has no contrary evidence and refuses a parentage test, the people's court may determine that the claim of the party requesting the confirmation of the maternity or paternity is tenable.

The effectiveness of the paternity presumption is demonstrated by the opponent's ability to refute the alleged facts. The presumption of legitimacy may lead to the legal father being different from the biological father, with the legal father assuming parental responsibility for the child's upbringing, education, and protection. Article 39 of Interpretation I of the Marriage and Family Book, which states that "the other party has no contrary evidence," indicates that the presumption of paternity in China is rebuttable and that the defendant can rebut the presumption of paternity claimed by the plaintiff by presenting contrary evidence. In order to balance the interests of both parties, the adversarial rights of the defendant correspond to the plaintiff's burden of mitigating proof.

Therefore, when the plaintiff seeks relief under the burden of proof procedural standards, the defendant should be afforded the same degree of protection. The law provides adversarial rights to the defendant in order to circumvent the plaintiff's and parties' procedural rights due to excessive tilt. In order to prevent an imbalance of interests between the parties resulting from the plaintiff's procedural rights, the law grants the defendant the right to confront. Consequently, the law empowers the defendant to refute the inferred facts by allowing the defendant to present contradictory evidence during the proceedings. If the alleged facts are refuted, the plaintiff must present the evidence again or face the penalties, i.e., the burden of proof lies with the plaintiff.

## 5. Acknowledgment of Parent-Child Relationship: Claim

A claim is a legal method for establishing the legitimacy of children born outside of wedlock. It constitutes a comprehensive parent-child legal system with additional regulations of legitimacy presumption, denial of maternity or paternity, affirmative, etc. Nonetheless, the Civil Code of the People's Republic of China regulates claims only in principle, as opposed to providing specific rules such as facultative claim, mandatory claim, denial of maternity or paternity, and positive.

### 5.1. The Provisions of the Claim Are Unclear

In essence, the institution of recognition alters the status of the child. The biological father acknowledges that the child born outside of marriage is his own, thereby establishing the child's legitimacy. The biological father is the biological father, meaning the child is biological and related by blood to him. The biological father claims the illegitimate child through a paternity suit in court, and the court's determination as to whether this recognition is required to establish a blood relationship between the two parties differs. This discrepancy leaves the illegitimate child unsure of the identity of the biological father, which negatively impacts his parental rights to protection, education, and support.

According to Article 1073, paragraph 1, of the Chinese Civil Code, the recognition and claim of a biological child born outside of wedlock by the biological father as a legitimate child is essentially a claim of the child. However, this provision is silent on claim types,

claim denial, compulsory claims, and quasi-certification, indicating that China has not yet developed a legal system for claims.

China has made general provisions regarding claim, which must be supplemented by specific rules governing their application; otherwise, they may result in questionable decisions. The lack of clarity in the rules of recognition and the different factual conclusions reached by judges render the application of the rules of recognition more discretionary and result in inconsistent decisions. Therefore, it is essential that claim rules be specific.

The purpose of a paternity confirmation lawsuit is to establish a new parental relationship, not to alter the existing paternity. The formation of a new parent-child relationship will inevitably result in the elimination and transformation of the original parent-child relationship (Chen 2020a). The nature of the parentage confirmation of the lawsuit should be the formation of the lawsuit because it changes the existing parentage, whether in terms of legal purpose or legal consequences. In addition to exercising parental rights to children, education, upbringing, protection, etc., but also to assume the social responsibilities of parents, the result of the formation of the litigation affects everyone (Guo et al. 2021).

*5.2. Operational Rules for the Claim System*

The Chinese Civil Code makes no clear distinction between voluntary claim and compulsory claim, and the procedure for both is identical to the procedure for compulsory claim, which is subject to prosecution. A claim must be confirmed by a court in order to be valid, which means that the plaintiff must provide the "necessary evidence" to prove paternity in order for the claim to be upheld by the court. Thus, the claim of a child born outside of wedlock is contingent on the father and son's shared ancestry. A father who is related by blood may directly petition the court to identify the child's biological father.

The purpose of the compulsory claim is to protect both the interests of children born out of wedlock and the interests of the biological mother. In China, there are currently three main legal presumptions for the acquisition of legitimate children: children born on the basis of parental marriage, voluntary claim, and the child lives with a new spouse who his parents remarry. Other than that, they are illegitimate children. In most cases, the interests of children born out of wedlock are compromised because the father and the child have not formed a protective education and upbringing relationship, so the law protects children born out of wedlock through a system of compulsory claims, whereby the child and the parents may at any time bring an action in court to establish paternity with the parent who has not formed a protective education and upbringing relationship, so as to protect the interests of the child. If a child born out of wedlock or a biological mother fails to claim a biological father who is related by blood and should have done so, the child born out of wedlock or the biological mother may bring an action against the biological father, with the blood relationship between the father and the child being a necessary condition; if there is no blood relationship, the court may not uphold the action.

Even if there is a blood relationship, the claim can be revoked through the denial of the lawsuit; however, the biological father cannot claim revocation due to the absence of a blood relationship if the adoption was finalized as a result of error, fraud, or coercion. Otherwise, if the biological father cannot be identified, the child will be fatherless from birth, which is contrary to the natural law of family lineage and the legislative intent of establishing paternity via claim. Therefore, in practice, the failure of a claim may be based on a variety of grounds, such as the absence of a parental relationship, the protection of the child's identity, and the observance of the child's wishes, without the absence of a blood relationship being a necessary condition.

Quasi-positive is as important as a claim in parentage affirmation. The Chinese Civil Code addresses the quasi-positive of children born outside of marriage, but quasi-positive exists in Chinese civil law as a source of kinship law in the form of social order and ethical custom, instead of provision. There are numerous de facto marriages in Chinese social life, during which children are presumed to be born out of wedlock, and when a marriage is registered between the parties, the child is born in wedlock and does not need to be

confirmed by administrative or judicial authorities. This demonstrates that China has a de facto quasi-correction system, although it is not explicitly provided for in the Marriage and Family section.

Article 1071 of the civil code states that:

Children born out of wedlock enjoy the same rights as children born in wedlock. No organization or individual may harm or discriminate against them.

The natural father or the natural mother who does not have custody of his or her child born out of wedlock shall pay the child support of the minor child or the child of full age who is incapable of living on his or her own.

The fact that Article 1071 of the Civil Code recognizes that children born out of wedlock have the same legal status as legitimate children demonstrates that China does not discriminate against children born out of wedlock and that operational clarification of quasi-correction is unnecessary, as there are no fundamental differences or special requirements for China's paternity system. Unquestionably, China is in line with the global trend of legislation to protect the interests of children, and the interests of children born outside of marriage are protected in the same manner as those born within marriage (Prophecy Coles 2020).

## 6. Denial of Parent-Child Relationship: Denial of Legitimate Children

*6.1. The Scope of the Birth Confirmation Action Is Expanded to Include Adult Children*

6.1.1. Child Information Rights Protection

According to Article 1073, in addition to the husband and wife, children also have the right of action for wrongful denial. However, the overall design is unable to embody the core value of the legal system because the principled provisions merely indicate the direction rather than enact it. However, if there are no stated conditions for implementation, the emphasis of the Civil Code may remain theoretical (Yang 2020). As a result, it is necessary to enhance interpretation and maintain consistency in law enforcement.

In recent years, the parent-child relationship legal framework of family law has encountered personality-related issues. Eventually, the focus shifted from the parent to the child. Consequently, when nations modify their laws, the concept of children's best interests is regarded as a crucial standard (Wang 2003). According to the growth of Chinese marriage and family laws, the litigation system of legitimate denial first appeared in the Civil Code introduced in 2021, and the objects of the case were limited to the father and mother. On the other side, it broadens the scope of the birth affirmation action to include the father, mother, and adult children.

6.1.2. Prevention of Rights Abuse

The second paragraph of Article 1073 of the Chinese Civil Code grants adult children the right to file a lawsuit on their own, but this right is limited to establishing the parent-child relationship and precludes contesting it. To confirm the parent-child relationship, adult children must be the subject of the action. The foundation of the right of children to know their blood relatives is the protection of their right to information.

In order to prevent adult children from evading their maintenance obligations to their parents by denying paternity, the Chinese legislator restricted adult children to taking actions to confirm paternity but not to deny paternity Ibid. (You 2021).

If the law stipulates that the obligations of parents and children are the same, then minors may enjoy rights equivalent to those of their parents. As adults, they may bring an action to deny the existing parentage if they have located their biological parents, but they cannot refuse to support the parent who raised them even if the parentage is dissolved. These measures adequately protect not only the rights of the elderly, but also the right of the parents to terminate paternity in this manner, preventing rights abuses while pursuing the authenticity of the blood relationship and the child's best interests.

In addition to prohibiting adult children from evading their support obligations through the parent-child connection denial legal system, the topic of confirming the parent-child relationship is restricted to adult children; young children cannot be the subject of litigation. In terms of children's best interests, the most important aspect of their adolescence is ensuring they receive family support and care. After reaching adulthood, family safety becomes less important. If the law permits children to know their actual consanguinity to identify their biological father, there will be minimal harm to their best interests. Instead of emphasizing that children have the option to change the legal parent-child relationship, the right of children to know their blood relationship should be limited to self-protection (S. Zhou 2012).

*6.2. A Summary of the Legal Rejection Lawsuit's Subject Matter*

According to Chinese civil procedure law, excluding the heir from the subject of the rejection litigation is consistent with the style of the current procedural law. In an identification case, only the directly interested parties, including the presumed father (husband), biological mother (wife), and children, have the right to contest the legal status of the children. If the husband or wife dies during or before the legal prosecution period and the law permits his heirs to file a case under the extra lawsuit for denial of maternity or paternity, it is in breach of the Civil Code's restrictions on the scope of the subject matter of the proceedings (Guo 2009).

In addition, there is a conflict of interest between the heirs and the inheritance. Even if the court holds a trial, it must speculate on the parties' preferences and conclude that, if the husband or wife of the family member dies prior to filing the complaint for denial of maternity or paternity, normal living cannot be anticipated between their parents and children, and the family security on which they rely will be lost. Therefore, it is unnecessary to investigate blood relations. Outside heirs may file a case for the denial of illegitimate birth, excluding the parent-child relationship suggested by legal birth (Jiang 2015). In short, Article 1073 of the Chinese Civil Code firmly restricts the subject of the lawsuit to the confirmation (or denial) of actual birth, which is consistent with substantive and procedural law.

The original legal basis for children as the subject of litigation regarding the confirmation or denial of legitimate birth can be found in the United Nations Convention on the Rights of the Child and Germany's original family laws, which can be found in Article 1073 of the Chinese Civil Code. Under Chinese civil law, adult children can only seek confirmation of their birth through marriage. This regulation prohibits adult children from not meeting their child support obligations.

If their interests are violated when they are minors, such as having no real family life or being mistreated by their parents, and the husband and wife are unwilling or unable to sue for denial of maternity or paternity, can the children sue on their own behalf to terminate the parent-child relationship? If minors may sue, then so may their legal guardians (father or mother). As an interested party, the father or mother creates a conflict that makes it difficult for the children's perspectives to be heard equally. Therefore, if there is no compelling need for family protection and the biological father wants to claim the child, he may submit a petition for denial of pregnancy or paternity (Chen 2015).

The protection of the best interests of the child is a fundamental principle in the Marriage and Family Book of the Chinese Civil Code, which is derived from the UN Convention on the Rights of the Child. As is well known, Article 3 of the Convention on the Rights of the Child establishes the best interests of the child as a fundamental principle of child protection, and best interests is increasingly becoming an authoritative term in the field of child protection. However, this article does not explain the meaning and criteria of the principle of best interests, and the relationship between this criterion and social traditions has always been a problem, and the flexibility and ambiguity of the principle have made its implementation difficult. In the course of the development of the best interests principle, its connotation and extension have been challenged by traditional

practice as well as by theories. The international community has, of course, made efforts to achieve a relatively certain meaning of the principle of the best interests of the child in its implementation.

To accommodate the trend of protecting children's rights, the Chinese Civil Code has expanded the issue of legitimate confirmation litigation to include adult children in addition to husband and wife. Additionally, the absence of the biological father and heir has legal implications. In accordance with the principle of preserving marriage and family, it is inappropriate for the birth father to invade the family or file a paternity or pregnancy denial suit. In contrast to German family law, Chinese family law places a greater emphasis on the substance of marriage and the family. Consequently, a limited legal opening can be created for the biological father's right of action.

## 7. Legal Drafting of Paternity in the Chinese Civil Code: Adoption

Adoption is a legal act in which a natural person adopts another person's child as his own through legal procedures, thereby establishing a legally mimetic parent-child connection between the adopter and the adoptee. When adoption creates a relationship between adoptive parents and children, their rights and obligations become a pseudo-blood relative relationship between the adopter and the adoptee. Adoption is one of the civil code's primary systems, and adoptive parents and children are an important type of parent-child relationship and one of the fundamental contents of kinship. The Civil Code has amended the adoption system to bring it into line with the basic principles of marriage and family law, changes in the three-child birth policy, and the development of social ideas.

The transition of adoption law into Volume V is one of the most significant alterations. When the Civil Law went into effect, the three chapters of the previous adoption law, General Provisions, Legal Liability, and Supplementary Provisions, were repealed and replaced by the applicable provisions of the Civil Code (including other chapters of the code). In addition, to protect the legitimate rights and interests of adoptees, the principle from the general provisions of the old adoption law that "adoption should be conducive to the upbringing and development of adopted minors" has been elevated to the General Rules chapter of the Book of Marriage and Family. Adoption must be based on the principle of serving the best interests of the adopted individual while also protecting the rights and interests of both the adopter and the adopted individual (Article 1044). As a fundamental principle of adoption, it incorporates the concept of best for children from the Convention on the Rights of the Child, the Chinese Constitution, and family laws.

Another characteristic of the adoption chapter is the modification and improvement of the conditions for the establishment of adoption (Chapter V, Section I of the Marriage and Family Book), and the modifications in the events for the establishment of adoption are as follows.

### 7.1. Relax the Standards for Substantial Adoption

First, the 14-year-old age restriction has been eliminated, permitting the adoption of minors (i.e., individuals under the age of 18) who meet the legal requirements (Article 1093). It is necessary to broaden the scope of adoptees to prevent the occurrence of minors over the age of 14 (including 14 years of age) who are unable to support themselves for a variety of reasons, such as the death of their parents, the inability to identify their biological parents, and the hardship on their biological parents, but who cannot be adopted. Protecting the interests of individuals who are in demand and qualified is facilitated by expanding the pool of minor adoptees.

Second, regardless of whether or not an adoptive parent has previously raised a child, adoptive parents can now adopt a second child (Article 1098, paragraph 1). By removing the restriction on the number of children raised through adoption, this law has actually increased the number of adopters.

The number of children adopted by adoptive parents has finally increased. Therefore, childless adoptive parents can adopt two children, while adoptive parents with one child

can only adopt one (Article 1100). This number is in accordance with China's Population and Family Planning Law, which stipulates that a couple may foster no more than three children within a nuclear family.

### 7.2. Increase the Restrictions on the Adoption of Bachelordom

Adoption by a single parent should foster healthy child development and adhere to the principle of gender equality. In order to prevent ethical violations and protect the interests of minor women, Article 9 of the old adoption law stipulated that if a single man adopts a girl, the age gap between them must be greater than 40 years. While the civil code changes this to when a single person intends to adopt a child of the opposite gender, the adopter must be at least 40 years older than the child (Article 1102). It indicates that the original gender discrimination laws were modified. This is to prevent single adopters, regardless of gender, from engaging in illegal activities through adoption.

### 7.3. Make the Adoption Requirements More Strict

Article 1098 now includes an extra criterion for adopters as Item 4. There are no unlawful or criminal histories for the adopter, which might affect the adoptee's development.

The law requires adopters to be of good moral character to avoid committing acts that may endanger the physical and emotional health of children, such as abuse, abandonment, sexual offenses, violations, and other criminal activities.

### 7.4. Include the Adoption Evaluation

According to the code, the department of civil affairs of the people's government at or above the county level is responsible for evaluating the adoption acts of the law (Article 1105, item 5). Departments of civil affairs or designated organizations must conduct a scientific evaluation of the adopter's motivations, health status, moral conduct, financial circumstances, marriage and family status, and family members' readiness to live together. To ensure the healthy development of adopted children, the adoption registration authority shall register adopters with favorable conditions and abilities based on the adoption assessment report produced by the assessment institution.

In conclusion, the adoption establishment requirement amendment protects adoptees and prevents child trafficking under the guise of adoption. It is also concerned with striking a balance between the interests of adopters and adoptees, promoting family-based old-age care and childrearing, and safeguarding the legitimate rights and interests of adopters. In addition, the current amendment of the requirements for adoption establishment takes into account changes in China's family planning policies and regulations, which now permit the third child to be raised in each family, as opposed to the previous policies of "one-child" and "late marriage childish." When the requirements for establishing an adoption changed, so did the family planning regulations.

## 8. Conclusion and Further Research

This article analyzes and outlines Chinese marriage and family law by investigating the parenthood legal provisions that highlight family laws theoretically. In terms of substance, the Marriage and Family Book has made enormous progress in improving the norms of family law in China and should be fully recognized; yet, there are still gaps in reasonably fundamental rules, a lack of explicit regulations, and a lack of some kinship legal systems. When applying the Marriage and Family Law in court, the legal requirements must be appropriately read, and marriage and family connections, i.e., kinship links, must be properly adjusted Ibid. (You 2021).

A comprehensive paternity system comprises the presumption of paternity, the acknowledgment of paternity, i.e., the claim system, and the denial of paternity, i.e., the act of denying legitimacy. Article 1073 of the Chinese Civil Code establishes a paternity recognition system for the first time. This article allows for the acknowledgment and denial of paternity and represents a significant advance in Chinese family law. The fact that it is a

code of principles that is insufficiently operational is, however, a significant issue. In order to make this provision operational, it must be interpreted and refined, which is the purpose of this article.

As an essential component of the legal system of parentage, the rule of presumption of parentage is the premise and foundation of the entire parentage system. In China, there is both a legal and a factual presumption of parentage. In accordance with the legislative intent and objective expression of the legal provisions on parentage in the Civil Code, the legal attributes, legal logic, and legal effects of the presumption of parentage are clarified, and deviations in the application of the law resulting from inconsistent understanding of the law in judicial practice are positioned and corrected. The presumption of paternity rule focuses on the determination of paternity, as maternity can be established based on the fact of birth, whereas paternity can only be established on the basis of a presumption. When paternity is incorrectly presumed, the plaintiff is required to refute the assumption through a denial of legitimacy claim.

The system of claim is the way for illegitimate children to be legitimate children, even though there is no substantial difference between them in Chinese civil laws. The reason is that the principle of best for children plays an important role in the adoption laws of the Chinese Civil Code. Children born out of wedlock and children born in wedlock have the same legal status in the Chinese Civil Code, and there is no difference in their rights and obligations, only that they are referred to differently. There are some historical reasons for the lack of an explicit recognition system in Chinese kinship law, but there have always been practical norms of recognition and quasi-correction in Chinese justice. It is just that the Chinese Civil Code has not yet developed a system of claim in legislation, which, coupled with the inconsistent determination by the courts of the necessity of the genuineness of blood, may lead to contradictory decisions. Thus, this paper refines the general rules on the recognition of filiation as stipulated in Article 1073 of the Civil Code and adds operational rules on the manner and elements of claim, the period of claim litigation, and quasi-positive.

Article 1073 of the Civil Code and its judicial interpretation is also a relief mechanism for the presumption of paternity, but in order for this article to have the desired legal effect, the subject of the lawsuit for denial of legitimacy, the grounds for denial, and the period of prosecution should be refined. For the law to permeate the "claim" system, it should also be supplemented by rules, unified application, so that it is adapted to the Chinese legal system to ensure that the paternity or denial of the action is operable. The litigation subjects of the denial lawsuit of out-of-wedlock are limited to the father and mother, and he or she is asked to provide proof that they have good cause for the litigation. The aim of this system is to strike a balance among the blood creek confederacy, the family status insecurity, and the best interests for children. The essence of the parent-child relationship system is to accept or reject the factors that influence the affirmation of parenthood by the legal design, and as a result, the legal framework can achieve justice.

Adoption is a legal process that proposes a new set of parents to a child. On the other hand, adoption does not require a preexisting biological bond between the adoptive parents and the child. Therefore, in order to safeguard the interests of the adopted person once a parent-child relationship has been established with the adopter, stringent restrictions should be imposed on the conditions of the adopter.

The Chinese Civil Code establishes the Chinese parentage law system and contains provisions for the recognition of parentage, which represents a significant advance in the law governing parental relationships. Even though this article is a norm of principle and there are no operational rules, the development of Chinese parentage law is undeniable. Respecting the private nature of kinship law, maintaining the kinship legal system, supplementing the main legal provisions with jurisprudential precedent, adapting to the needs of society and the times, meeting the legitimate expectations of society and the parties, ensuring the correct implementation of the marriage and family codification, and preserving the stability of marriage and the family are therefore essential when applying the law.

**Funding:** This research received no external funding.

**Institutional Review Board Statement:** Not applicable.

**Informed Consent Statement:** Not applicable.

**Data Availability Statement:** Data sharing does not apply to this article as no datasets were generated or analyzed during the current study.

**Conflicts of Interest:** The author declares no conflict of interest.

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
