# Peer review of "Parent-Child Relationship in the Civil Code of China"

_laws, 2022_

Round 1

Reviewer 1 Report

The article deals with very interesting and extremely current topics. In fact, not only the Civil Code of the People's Republic, which regulates marriage and family in its fifth book, entered into force on 1 January 2021. The Chinese family, which has always been regarded as the basic cell of society, has also undergone important changes because of economic reforms, and in recent years has acquired an increasingly important political role. Therefore, a study on the evolution of Chinese family law, with particular reference to the parent-child relationship, would be vastly useful, and provide an advance in the knowledge of the Chinese socio-legal landscape. Unfortunately, the article is rather tiring to read, both for formal reasons and structure issues. With regard to the first aspect, in addition to some typing errors there are terminological inaccuracies or unclear expressions (e.g. line 318: what is a ‘program effect’? how does it contrast with the ‘legal effect’ in line 337?; line 82: what is the 'draft of expert counsel on parentage laws’?; lines 364-367; lines 46 – 66; lines70-76). The structure of the article is also not entirely consistent with what is stated in the abstract. In particular, the article does not answer some general questions concerning the characteristics of Chinese family law (lines 6-9), focusing instead mainly on the analysis of Article 1073, with the result of being sometimes repetitive, and more often obscure. It is also difficult to understand how the section on adoption fits into the rest of the discourse on the parent-child relationship. Finally, the conclusions do not seem entirely consistent with what was analysed in the paper.

I, therefore, recommend an accurate linguistic revision, failing which I would be inclined to consider the paper not worthy of publication. As for the content, I would advise adding some references to law in action. Moreover, I would focus more on the structure, trying to get a better understanding of the logical connection between the various parts. Then, I would revise the abstract and the conclusions to make them more coherent with the topics covered. I would also suggest paying more attention to the footnotes, and adding references when quoting the thoughts of some authors (see e.g. lines. 136-138).

Author Response

Reply to reviewer

Manuscript ID: ISSN 2075-471XTitle: Parent-child Relationship in the New Civil Code of China

Authors: Wenting You

Thank you very much for the valuable suggestions. The issues raised were of great help in our revision. In the past days, I have made extensive modifications on the original manuscript based on your comment and request. In the following pages are my point-by-point responses to each of the comments of the both reviewers.

1.The article would benefit from a more critical perspective of family law concepts that go beyond the Chinese context. There should be more discussion of the United Nations Convention on the Rights of the Child. While it is briefly mentioned in the article, concepts such as the best interests of the child principle should be considered in analysing whether the Chinese civil code is focused on promoting the child's best interests as paramount in the assessment of the status of the child in legal parentage cases.

Reply:  The protection of the best interests of the child is a fundamental principle in the Marriage and Family Book of the Chinese Civil Code, which is derived from the UN Convention on the Rights of the Child. As is well known, Article 3 of the Convention on the Rights of the Child establishes the best interests of the child as a fundamental principle of child protection, and Best Interests is increasingly becoming the authoritative word in the field of child protection. However, this article does not explain the meaning and criteria of the principle of best interests, and the relationship between this criterion and social traditions has always been a problem, and the flexibility and ambiguity of the principle has made its implementation difficult. In the course of the development of the best interests principle, its connotation and extension have been challenged by traditional practice as well as by theories. The international community has, of course, made efforts to achieve a relatively certain meaning of the principle of the best interests of the child in its implementation. (L550-562)

  1. There could also be discussion on the principle of mater semper certa est. This refers to the concept that "the mother is always certain" in cases involving the status of the child in legal parentage cases. In many other countries around the world, the birth mother is presumed to be the legal mother and the birth mother's spouse or partner is considered to be the legal father. This should be considered and compared with China's legal approach. If these issues were to be considered in more critical detail, it would help lift some of the descriptive parts of the article into more critical analysis.

Reply: Chinese law is in line with other national legislation in terms of the identification of motherhood. The legislation of most civil law countries follows the constant principle of Roman law that "the one who gives birth is the mother", and that the fact of birth automatically gives rise to motherhood. As a rule, the legal relationship between the child and the biological mother is not easily disputed and motherhood is automatically acquired on the basis of the fact of the child's birth or the registration of the mother's name on the birth certificate. The acquisition of paternity or maternity is linked to the existence of a marital relationship and the fact of the birth of the child. The presumption of motherhood is generally based on the fact of birth, registration or possession of identity. In addition, motherhood can also be established by possession of civil status. Take French Law for example, Article 311-1 of the French Civil Code provides that, in addition to the fact of birth, motherhood may also be established by possession. (L161-172)

  1. The abstract contains too many rhetorical questions. It is best to simply describe what you will be arguing rather than making the reader guess and form conjectures about the article.

Reply: The purpose of this article is to familiarize readers with the Chinese Civil Code, which entered into force in early 2021, and to draw their attention to the changes brought about by the Marriage and Family Book, which is now included in Volume V of the new code. The paternity system best reflects the changes in Chinese Marriage and Family Book, especially the article 1073. A complete paternity system includes presumption, claim and denial of parent-child relationship. However, article 1073 of the Civil Code, which regulates parent-child relationship, is a guiding provision with the lack of operational rules. It is necessary to make general rules for operation and enforceable by adding supporting rules, including the presumption of legitimate children, the claim of children born out of wedlock, the denial of legitimate children, and other operational rules, to resolve paternity disputes. The Civil Code also makes changes to the adoption system in the Marriage and Family Book, mainly by further restricting the conditions for adopters, with the aim of protecting the interests of the adoptee children. Although the Chinese Civil Code retains the concepts of legitimate and illegitimate children, in essence there is no difference in their rights and legal status, including the right to inheritance. In conclusion, the legislative norms of paternity determination improve the Chinese paternity system, but lack operability, and it is important to accumulate experience through practice and draw on custom and jurisprudence to develop specific operational rules that complement the legislative provisions. This is exactly what this paper will address and the knowledge gap to fill. (L3-20)

  1. There could also be further discussion about illegitimacy of children and its consequences in the law of succession.

Reply: Quasi-positive is as important as claim in parentage affirmation. The Chinese Civil Code addresses the quasi-positive of children born outside of marriage, but quasi-positive exists in Chinese civil law as a source of kinship law in the form of social order and ethical custom, instead of provision. There are numerous de facto marriages in Chinese social life, during which children are presumed to be born out of wedlock, and when a marriage is registered between the parties, the child is born in wedlock and does not need to be confirmed by administrative or judicial authorities. This demonstrates that China has a de facto quasi-correction system, although it is not explicitly provided for in the Marriage and Family section.

Article 1071 of the civil code states that:

Children born out of wedlock enjoy the same rights as children born in wedlock. No organization or individual may harm or discriminate against them.

The natural father or the natural mother who does not have custody of his or her child born out of wedlock shall pay the child support of the minor child or the child of full age who is incapable of living on his or her own. The fact that Article 1071 of the Civil Code recognizes that children born out of wedlock have the same legal status as legitimate children demonstrates that China does not discriminate against children born out of wedlock and that operational clarification of quasi-correction is unnecessary, as there are no fundamental differences or special re-quirements for China's paternity system. Unquestionably, China is in line with the global trend of legislation to protect the interests of children, and the interests of children born outside of marriage are protected in the same manner as those born within marriage. (L453-474)

  1. There are also missing citations for the Old Roman phrase "family orientation" (you also need to state the Latin expression) on page 2 and the quote from Engels on page 3.

Reply: L154, L53 Add the two citations.

  1. On page 9, the author states "Denial of the parent-child relationship: Denial of legitimate children." This is not a proper English sentence because it is missing a verb that connects these two clauses. Please rewrite this sentence.

Reply: This incorrect sentence has been delete.

  1. Also on page 9, there is a sentence: "Adoptive parents who adopt minors are so saints and good." This does not make sense in English, please rewrite this sentence too.

Reply: Adoption, a legal process that proposes a new set of parents to a child. Adoption, on the other hand, does not require a preexisting biological bond between the adoptive parents and the child. Therefore, in order to safeguard the interests of the adopted person once a parent-child relationship has been established with the adopter, stringent restrictions should be imposed on the conditions of the adopter. (L702-706)

  1. The conclusion is also too short and it needs to be longer to elaborate on the shortcomings of the law that are mentioned but not explained in enough detail.

There also appears to be some formatting errors in the list of references that need to be corrected in the editing stages.

Reply: The “Conclusion and further research” part has been rewritten in the line 650 to 716.

Reviewer 2 Report

The article analyses changes to parenting and adoption law in the Chinese Civil Code that have been in operation since 2021. Overall the article is well written with a clear and thoughtful consideration about legal parentage and adoption law. Parts of the article could be further explained in more detail. Moreover, there are some English writing issues that need to corrected before the article can be published. Therefore, the article should be provisionally accepted based on the academic contribution its makes, but this is subject to minor revisions, including correcting grammatical errors and improvement of analysis in some areas.

The article is novel based on its analysis of recent legal changes in the Chinese Civil Code that warrants academic discussion. The author primarily relies on Chinese sources, which is appropriate for research on Chinese law. However, the article would benefit from a more critical perspective of family law concepts that go beyond the Chinese context. There should be more discussion of the United Nations Convention on the Rights of the Child. While it is briefly mentioned in the article, concepts such as the best interests of the child principle should be considered in analysing whether the Chinese civil code is focused on promoting the child's best interests as paramount in the assessment of the status of the child in legal parentage cases. There could also be discussion on the principle of mater semper certa est. This refers to the concept that "the mother is always certain" in cases involving the status of the child in legal parentage cases. In many other countries around the world, the birth mother is presumed to be the legal mother and the birth mother's spouse or partner is considered to be the legal father. This should be considered and compared with China's legal approach. If these issues were to be considered in more critical detail, it would help lift some of the descriptive parts of the article into more critical analysis.

The abstract contains too many rhetorical questions. It is best to simply describe what you will be arguing rather than making the reader guess and form conjectures about the article. There could also be further discussion about illegitimacy of children and its consequences in the law of succession.

There are also missing citations for the Old Roman phrase "family orientation" (you also need to state the Latin expression) on page 2 and the quote from Engels on page 3. On page 9, the author states "Denial of the parent-child relationship: Denial of legitimate children." This is not a proper English sentence because it is missing a verb that connects these two clauses. Please rewrite this sentence. Also on page 9, there is a sentence: "Adoptive parents who adopt minors are so saints and good." This does not make sense in English, please rewrite this sentence too.

The conclusion is also too short and it needs to be longer to elaborate on the shortcomings of the law that are mentioned but not explained in enough detail. There also appears to be some formatting errors in the list of references that need to be corrected in the editing stages.

Despite the above mentioned shortcomings, the article nevertheless still has merit in the analysis of the Chinese family law. There are some interesting observations about the changes to the Chinese law on parentage and adoption that would be interesting to a wider audience. The article ought to be revised with the above mentioned points in mind in order to transform the article into a stronger and publishable piece of scholarship.

Author Response

Reply to reviewer

Manuscript ID: ISSN 2075-471XTitle: Parent-child Relationship in the New Civil Code of China

Authors: Wenting You

Thank you very much for the valuable suggestions. The issues raised were of great help in our revision. In the past days, I have made extensive modifications on the original manuscript based on your comment and request. In the following pages are my point-by-point responses to each of the comments of the both reviewers.

  1. The article is rather tiring to read, both for formal reasons and structure issues. With regard to the first aspect, in addition to some typing errors there are terminological inaccuracies or unclear expressions. The structure of the article is also not entirely consistent with what is stated in the abstract. Moreover, I would focus more on the structure, trying to get a better understanding of the logical connection between the various parts.

Reply:  The English expression and the structure has been modified in the article as colorful words.

  1. In particular, the article does not answer some general questions concerning the characteristics of Chinese family law (lines 6-9), Delete focusing instead mainly on the analysis of Article 1073, with the result of being sometimes repetitive, and more often obscure.

Reply: The abstract and the conclusion has been rewritten with a responding of the main body of the article. (L3-20) Adoption is a legal drafting parent-child relationship which is stipulated in the Volume V and made some amendment. So I have explore the system in the code.

  1. It is also difficult to understand how the section on adoption fits into the rest of the discourse on the parent-child relationship.

Reply: It is the claim system of the parentage in Chinese civil code, but not adoption. I have add the paragraphs about the claim. (L387-474)

  1. Finally, the conclusions do not seem entirely consistent with what was analysed in the paper.

Reply: The conclusion has been rewritten to correspond the main body of the article. (L650-716)

  1. As for the content, I would advise adding some references to law in action.

Reply: I have add the references to law and the statements in action.

  1. I would revise the abstract and the conclusions to make them more coherent with the topics covered.

Reply: The abstract and the conclusion have been written as the main body of article has been modified.

  1. I would also suggest paying more attention to the footnotes, and adding references when quoting the thoughts of some authors.

Reply: The two citations has been added in the footnotes.

Round 2

Reviewer 1 Report

The article has been significantly improved. I still recommend a careful reading to eliminate the last linguistic inaccuracies (e.g. l. 26-28: For China, this is a continuation of the civil law heritage, same as Swiss, German, and Japanese laws, arrangement ???? by the code's structure). After that, I think it can be published.

Author Response

I have written the sentence in line 26-28 in green color. Please see the attachment.
